

# Effects of sports intervention on aggression in children and adolescents: a systematic review and meta-analysis

Yahui Yang[1], Hao Zhu[2], Kequn Chu[1], Yue Zheng[3] and Fengshu Zhu[1]

[1] Yangzhou University, Yangzhou, Jiangsu, China
[2] Shanghai University of Sport, Shanghai, China
[3] Jiangsu Vocational and Technical College of Economics and Trade, Nanjing, Jiangsu, China

## ABSTRACT

**Objective**. To explore the impact of sports on aggression in children and adolescents and analyze whether different conditions in the intervention, such as type of sports, or intervention duration, have different influences on the effect of interventions.

**Method**. The study protocol was registered in PROSPERO (CRD42022361024). We performed a systematic search of Pubmed, Web of Science, Cochrane library, Embase and Scopus databases from database inception to 12 October 2022 for all studies written in English. Studies were included if they met the following PICO criteria. All analyses were carried out using the Review Manager 5.3 Software. We summarized aggression, hostility and anger scores using SMDs. Summary estimates with 95% confidence intervals were pooled using DerSimonian-Laird random effects model or fixed effects model according to between-study heterogeneity.

**Results**. A total of 15 studies were deemed eligible for inclusion in this review. The overall mean effect size indicated that sport interventions was associated with lower aggression (SMD $= -0.37$, 95% CI [$-0.69$ to $-0.06$], $P = 0.020$; $I^2 = 88\%$). Subgroup analyses showed that non-contact sports were associated with lower aggression (SMD $= -0.65$, 95% CI [$-1.17$ to $-0.13$], $P = 0.020$; $I^2 = 92\%$) but high-contact sports were not (SMD $= -0.15$, 95% CI [$-0.55$ to $0.25$], $P = 0.470$; $I^2 = 79\%$). In addition, when intervention duration $<6$ months, sport interventions was associated with lower aggression (SMD $= -0.99$, 95% CI [$-1.73$ to $-0.26$], $P = 0.008$; $I^2 = 90\%$) and when intervention duration $\geq 6$ months, sport interventions was not associated with lower aggression (SMD $= -0.08$, 95% CI [$-0.44$ to $-0.28$], $P = 0.660$; $I^2 = 87\%$).

**Conclusion**. This review confirmed that sports intervention can reduce the aggression of children and adolescents. We suggested that schools can organize young people to participate in low-level, non-contact sports to reduce the occurrence of bullying, violence and other aggression-related adverse events. Additional studies are needed to determine which other variables are associated with aggression in children and adolescents, in order to develop a more detailed and comprehensive intervention programme to reduce their aggression.

Corresponding author
Fengshu Zhu, fszhu@yzu.edu.cn

## INTRODUCTION

Aggressive behavior (AB) is defined as acts that directly target others with the intention of causing immediate harm to others, such as violence and bullying (*Anderson & Bushman, 2002*; *Azimi, Vaziri & Kashani, 2012*). A study reported that about 51% adolescents showed a high level of aggression in secondary school, and the aggressive tendency indicated a significant growth trend throughout adolescence (*Hamza et al., 2019*). Adolescents aggression exert a negative impact on perpetrators, victims and bystanders in varying degrees (*Wolke & Lereya, 2015*). Bullying in adolescence increased the risks of poor academic performance, poor school adjustment, substance abuse, and violent and criminal behavior in later life (*Moore et al., 2017*; *Schoeler et al., 2018*). Aggression would not only lead the implementers to develop internalized emotional problems and externalized problem behaviors, but also bring serious psychological adaptation problems to the victims (*Troop-Gordon, 2017*). As a result of bullying, victims suffered adverse mental health, physical, and psychosomatic problems such as depression, suicide, stomach aches, and insomnia (*Moore et al., 2017*; *Schoeler et al., 2018*). There were also psychological and behavioral problems reported by bystanders, such as anxiety, interpersonal sensitivity, and fears of further victimization (*Rivers et al., 2009*). Aggression had seriously affected the physical and mental health, academic progress, personality development and social adaptation of adolescents (*Gini & Pozzoli, 2013*; *Gini et al., 2014*).

At present, many studies have confirmed that sport is inversely associated with adolescent violence. The energy can be released by venting people's aggressive impulse in an appropriate way, so as to eliminate the aggressive tendency. Regular participation in sports could reduce the aggression of young people, because it provided frequent energy release opportunities (*Karin, Daniel & Roland, 2010*). Sports intervention has a positive effect on aggressive behavior of children and adolescents (*Kim, 2016*). The higher the physical activity level of school-age children, the lower their aggressive behavior (*Pino-Juste, Portela-Pino & Soto-Carballo, 2019*). *Fung & Lee (2018)* found that Chinese martial arts can effectively reduce the reactive and proactive aggression of school-age children. Sports can help reduce adolescent aggression. For example, after-school volleyball program may reduce aggressive behavior of adolescents by adjusting fun, motivation and self-control (*Trajković et al., 2020a*). Participating in organized school sports can strengthen teenagers' sense of belonging and dependence on school, and these characteristics will guide them to create and maintain a positive and orderly school environment, so as to stay away from violent and destructive acts (*Smith, 2011*).

However, not all studies have found a negative relationship between sports and adolescent violence. A meta-analysis reported that there was no overall significant association between sports participation and juvenile delinquency, sports participation could not reduce the occurrence of juvenile delinquency (*Spruit et al., 2016*). *Méndez, Ruiz-Esteban & Ortega (2019)* indicated that students who practiced physical activity at least four or more times per week, had higher values in the indicators of aggressiveness than students who practiced less frequently. *Mutz & Baur (2009)* reported that some rough physical contact in sports, or even fighting, actually leads to an increase in adolescent

aggression. *Kreager (2007)* found that high-contact sports such as football and wrestling led to increased violence, while non-contact sports such as baseball and tennis did not. *Zurita-Ortega et al. (2015)* reported that the overt aggressiveness of teenagers who practiced sport regularly was higher than sedentary teenagers, because they began to compete with each other.

This systematic review aims to integrate the existing research on sports intervention and explore the impact of exercise on children and adolescents' aggression. According to existing research, analyze whether different conditions in the intervention, such as type of sports, intervention duration, have different influence on the effect of interventions.

## METHODS

The study protocol was registered in PROSPERO (CRD42022361024).

### Search strategy

We performed a systematic search of Pubmed, Web of Science, Cochrane library, Embase and Scopus databases from database inception to 12 October 2022 for all studies written in English. The search strategy was designed by Yahui Yang and Fengshu Zhu by an initial scoping review of the literature. Studies were identified by using all possible combinations of the following groups of search terms: (a) "adolescent" OR "teens" OR "youth" OR "teenager" OR "juvenile" OR "young" OR "minor"; (b) "physical training" OR "sport" OR "exercise" OR "athletics"; (c) "intervention" OR "behaviour change" OR "prevention" OR "experiment" OR "program" OR "reduction" OR "evaluation" OR "strategy" OR "effect"; (d) "aggression" OR "bullying" OR "violence" OR "assaultive behavior" OR "atrocity" OR "physical assault" OR "fighting". The specific search was amended as necessary for each database to account for different search functionalities. The reference lists of retrieved articles and grey literature were searched to detect studies potentially eligible for inclusion.

### Inclusion and exclusion criteria

Studies were included if they met the following PICO criteria: (1) included typically developing children and/or adolescents (Population); (2) examined different sports including school physical education programs (Intervention); (3) included anactive/inactive comparator (Comparison) and (4) examined associations with aggression (Outcomes). Studies were excluded if they focused on populations with develop-mental disorders (*e.g.*, Down syndrome).

### Study selection

Search results were imported into Endnote to remove duplicates. Yahui Yang and Hao Zhu screened the titles and abstracts of the retrieved articles independently to remove irrelevant articles. Then the same reviewers independently screened remaining articles in full to determine the final included studies. Disagreements were resolved by consensus or consultation with Fengshu Zhu.

## Data extraction

One reviewer extracted specific characteristics from included studies, including country, study design, paticipants characteristics (age, gender), sample size, intervention programme characteristics (name, type, duration, frequency), comparison programme and outcome variables. *Shachar et al. (2016)* reported mean and standard deviation (SD) of baseline and change-from-baseline, the reviewer calculated the final mean and standard deviation according to Cochrane Handbook version 5.1.0 (*Higgins & Green, 2011*). Another reviewer confirmed the content.

## Outcomes

The primary outcome was aggression scores. The secondary outcomes were other externalizing behaviors of aggression, including hostility, anger, delinquent acts (including crimes of varying severity levels such as gang fights and extortion, but also minor theft and nuisances), attitude towards violence (ATV) and provocation/bullying scores. If outcomes were reported for more than one time point, we extracted results closest to post-intervention (*Fung & Lee, 2018*). If two or more measurement tools were used, we referred to a previously described hierarchy of outcome measures (*Fung & Lee, 2018*). If physical aggression and verbal aggression scores were reported concurrently, we extracted the physical aggression scores (*Trajković et al., 2020a*; *Trajković et al., 2020b*; *Shachar et al., 2016*; *Carraro, Gobbi & Moè, 2014*; *Reynes & Lorant, 2002*).

## Risk of bias assessment

Yahui Yang and Hao Zhu assessed risk of bias of randomised controlled trials (RCTs) using the Cochrane collaboration tool 2.0 (*Sterne et al., 2019*) and assessed risk of bias of quasi-experimental studies using the JBI Critical Appraisal Checklists for Quasi-Experimental Studies (*The Joanna Briggs Institute, 2016*) independently. Discrepancies were resolved by consensus or deliberation with Fengshu Zhu.

## Data analysis

All analyses were carried out using the Review Manager 5.3 Software. We summarised aggression, hostility and anger scores using SMDs. Summary estimates with 95% confidence intervals were pooled using DerSimonian-Laird random effects model or fixed effects model according to between-study heterogeneity (*DerSimonian & Laird, 1986*). The heterogeneity was estimated using $I^2$, considering $I^2$ values of $<25\%$, $25$–$50$, and $>50\%$ as small, medium, and large amounts of heterogeneity, respectively (*Higgins & Thompson, 2002*). Subgroup moderator analyses were conducted to determine whether results differed according to intervention duration and sport type. Sensitivity analyses were used to explore the impact of individual studies. A narrative synthesis of the results was carried out using descriptive statistics in order to summarize characteristics of the studies where data cannot be extracted (*Ioannidis, Patsopoulos & Rothstein, 2008*).

## RESULTS

### Study characteristics and risk of bias

Following the screening process, 15 studies (*Fung & Lee, 2018*; *Trajković et al., 2020a*; *Trajković et al., 2020b*; *Shachar et al., 2016*; *Carraro, Gobbi & Moè, 2014*; *Reynes & Lorant, 2002*; *Mehralian et al., 2022*; *Rosa et al., 2021*; *Harwood-Gross et al., 2021*; *Blomqvist, 2020*; *Wade et al., 2018*; *Setty, Subramanya & Mahadevan, 2017*; *Hortiguela, Gutierrez-Garcia & Hernando-Garijo, 2017*; *Park et al., 2017*; *Pels & Kleinert, 2016*) were deemed eligible for inclusion in this review. The detailed screening flow is shown in Fig. 1. Included studies were published between 2002 and 2022. Six studies were RCTs and nine studies were quasi-experimental studies. Twelve studies reported aggression outcomes, five studies reported hostility and anger outcomes, two studies reported delinquent acts outcomes and attitude towards violence, and one study reported provocation/bullying outcome. Three studies were comparison between two sports events with no control group. *Pels & Kleinert (2016)* carried out a single experiment and interventions of other studies varied in duration from 4 weeks to 1 year (see Table 1 for details).

Six RCTs and nine quasi-experimental studies were all identified as "moderate quality". The assessment results are shown in Tables 2 and 3.

### Impact of interventions on aggression

The overall mean effect size of nine included studies (*Fung & Lee, 2018*; *Trajković et al., 2020a*; *Trajković et al., 2020b*; *Shachar et al., 2016*; *Carraro, Gobbi & Moè, 2014*; *Reynes & Lorant, 2002*; *Mehralian et al., 2022*; *Wade et al., 2018*; *Park et al., 2017*) was SMD = −0.37, 95% Confidence Interval (CI) −0.69, −0.06, indicating that sport interventions reduced aggression compared to a control group ($P = 0.020$). There was significant heterogeneity between effect sizes between studies ($I^2 = 88\%$, $P < 0.001$) (Fig. 2).

According to the types of sport, non-contact sports (*Trajković et al., 2020a*; *Shachar et al., 2016*; *Mehralian et al., 2022*; *Wade et al., 2018*; *Park et al., 2017*) were associated with lower aggression (SMD = −0.65, 95% CI [−1.17 to −0.13], $P = 0.020$; $I^2 = 92\%$). High-contact sports (*Fung & Lee, 2018*; *Trajković et al., 2020b*; *Carraro, Gobbi & Moè, 2014*; *Reynes & Lorant, 2002*) were not associated with lower aggression (SMD = −0.15, 95% CI [−0.55 to 0.25], $P = 0.470$; $I^2 = 79\%$) (Table 4).

According to the intervention duration, when intervention duration <6 months (*Fung & Lee, 2018*; *Carraro, Gobbi & Moè, 2014*; *Mehralian et al., 2022*; *Park et al., 2017*), sport interventions was associated with lower aggression (SMD = −0.99, 95% CI [−1.73 to −0.26], $P = 0.008$; $I^2 = 90\%$). When intervention duration ≥6 months (*Trajković et al., 2020a*; *Trajković et al., 2020b*; *Shachar et al., 2016*; *Reynes & Lorant, 2002*; *Wade et al., 2018*), sport interventions was not associated with lower aggression (SMD = −0.08, 95% CI [−0.44 to −0.28], $P = 0.660$; $I^2 = 87\%$) (Table 4).

When the impact of individual studies was examined by removing studies from the analysis one at a time, we observed that when *Mehralian et al. (2022)*, *Park et al. (2017)*, *Shachar et al. (2016)* and *Carraro, Gobbi & Moè (2014)* been removed, the pooled results became insignificant ($P \geq 0.05$) (Table 5). However, these studies did not share any specific characteristics.
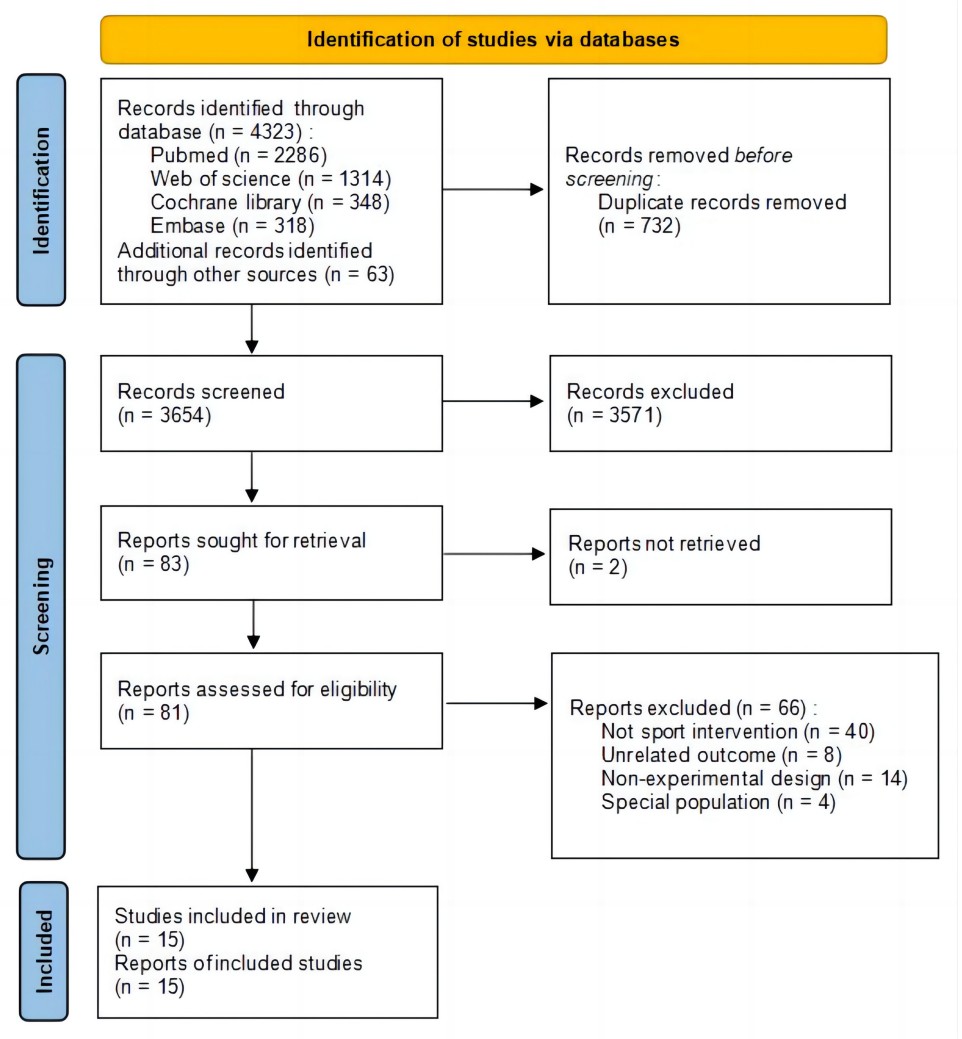

**Figure 1** Flowchart of the selection process.

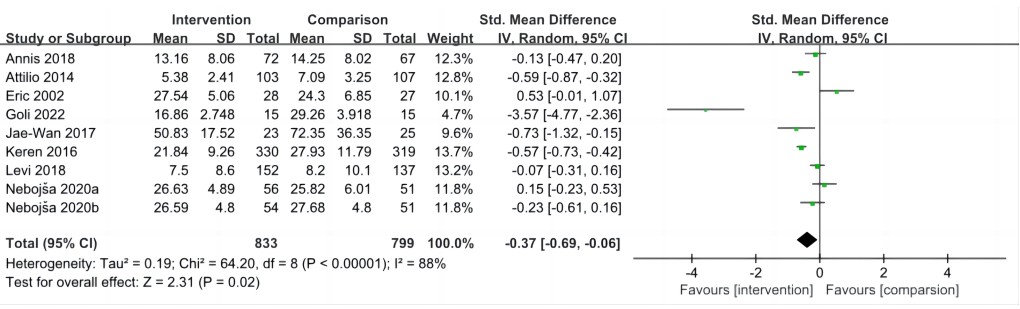

**Figure 2** Forest plot of studies for aggression.

**Table 1  Characteristics of included studies.**

| Study, Design, Country | Paticipants, Sample size, Age, Gender | Intervention programme | Comparison group | Intervention duration, frequency | Outcomes |
|---|---|---|---|---|---|
| *Mehralian et al. (2022)* | 7–10 year old girls | A child yoga-based mindfulness training package | No training | 10 one-hour training sessions | [a] |
| Quasi-experimental Study | Int. $n = 15$ (8 people aged 7–8 years and 7 people aged 9–10 years) | | | | |
| Iran | Con. $n = 15$ (8 people aged 7–8 years and 7 people aged 9–10 years) | | | | |
| *Rosa et al. (2021)* | Children and adolescents | ●Judo intervention | | 12 weeks, twice a week, lasting 60 min per session | [f] |
| Randomized Clinical Trial | Judo $n = 29$ (9.90 ± 1.56 years, 48% girls) | ●Ball games, including football, volleyball, basketball, and handball | / | | |
| Brazil | Ball games $n = 36$ (9.96 ± 1.51 years, 30% girls) | | | | |
| *Harwood-Gross et al. (2021)* | Boys from schools for at-risk youths, located in low socioeconomic areas | Martial arts classes | The same number of standard PE classes | 6 months, two 50-min classes per week | [a] |
| Quasi-experimental Study | Int. $n = 20$ | | | | |
| Israel | Con. $n = 19$ | | | | |
| | 15.6 ± 0.81 years | | | | |
| *Trajković et al. (2020a)* | Adolescents | Small-sided volleyball sessions and regular physical-education classes | Regular physical-education classes | 8 months, two scheduled 45-min sessions per week separated by at least 1 day | [abc] |
| RCT | Int. $n = 56$ (15.5 ± 0.7 years, 30% girls) | | | | |
| Serbia | Con. $n = 51$ (15.7 ± 0.6 years, 37% girls) | | | | |
| *Trajković et al. (2020b)* | High school students | Recreational soccer sessions and regular physical-education classes | Regular physical-education classes | 8 months, 64 sessions after school, 45-min sessions per week, separated by at least 1 day | [abc] |
| RCT | Int. $n = 54$ (15.7 ± 0.6 years, 26% girls) | | | | |
| Serbia | Con. $n = 51$ (15.8 ± 0.5 years, 31% girls) | | | | |
| *Blomqvist (2020)* | Students from local martial arts academies | ●Mixed Martial Arts (MMA) intervention | / | 5 months, at least twice a week | [ad] |
| Longitudinal Study | MMA $n = 63$ | ●Brazilian Jiu-Jitsu (BJJ) intervention | | | |

| Study, Design, Country | Paticipants, Sample size, Age, Gender | Intervention programme | Comparison group | Intervention duration, frequency | Outcomes |
|---|---|---|---|---|---|
| Sweden | BJJ $n = 50$<br>$20.23 \pm 2.43$ years<br>7% girls | | | | |
| *Wade et al. (2018)* | Boys in public, secondary schools located in low-income areas | ATLAS: a school-based, multicomponent physical activity program | No training | 8 months, continuous | [a] |
| RCT | Int. $n = 152$ | | | | |
| Australia | Con. $n = 137$<br>$12.7 \pm 0.5$ years | | | | |
| *Fung & Lee (2018)* | Children who scored $z \geq 1$ on the total score of the Reactive-Proactive Aggression Questionnaire | Wu gong (skill-based martial techniques): involved the basic hand-forms and foot stances, che quan (dragging punch), defense skills, and duichai (2-person combat sets) | The physical fitness training | 10 90-minute weekly sessions | [ad] |
| RCT | Int. $n = 72$ ($8.63 \pm 1.06$ years, 21% girls) | | | | |
| Hong Kong | Con. $n = 67$ ($8.57 \pm 1.11$ years, 32% girls) | | | | |
| *Setty, Subramanya & Mahadevan (2017)* | Children | Integrated yoga module | Moderate PE | 4 weeks, 1 h a day, 5 days a week | [e] |
| RCT | Int. $n = 76$ | | | | |
| India | Con. $n = 82$<br>12 years (13), 13 years (39), 14 years (36), 15 years (69), 16 years (1) 48% girls | | | | |
| *Hortiguela, Gutierrez-Garcia & Hernando-Garijo (2017)* | Students from fourth year of Secondary Education | Judo and capoeira teaching units | Football and basketball teaching units | 16/17 sessions | [e] |
| Quasi-experimental Study | Judo $n = 105$ | | | | |
| Spain | Ball games $n = 116$<br>$15.43 \pm 1.62$ years<br>51% girls | | | | |
| *Park et al. (2017)* | Children | Supervised progressive PEC program | No training | 8 weeks, continuous | [a] |
| Quasi-experimental Study | Int. $n = 23$ ($12.03 \pm 0.83$ years) | | | | |
| Korea | Con. $n = 25$ ($12.29 \pm 0.65$ years)<br>50% girls | | | | |

**Table 1** (*continued*)

| Study, Design, Country | Paticipants, Sample size, Age, Gender | Intervention programme | Comparison group | Intervention duration, frequency | Outcomes |
|---|---|---|---|---|---|
| *Pels & Kleinert (2016)* | Psychology or sport science students attending local universities | ●Rowing on an ergometer at a predefined pace of 12 kilometers per hour for five minutes | / | 10 min/6 min, once | [a] |
| Randomized Clinical Trial | Rowing $n = 30$ | ●A specific combat exercise for the duration of three minutes | | | |
| Germany | Combat $n = 30$ | | | | |
| | $24.05 \pm 3.31$ years | | | | |
| | 45% girls | | | | |
| *Shachar et al. (2016)* | Students had observed agressive behavior in Grades 3–6 | A total of 120 h of extra afterschool sports activities: comprising two weekly hours of martial arts and three weekly hours of other group sports activities | No training | 24 weeks, 5 h a week | [abc] |
| Quasi-experimental Study | Int. $n = 330$ | | | | |
| Israel | Con. $n = 319$ | | | | |
| | 24% girls | | | | |
| *Carraro, Gobbi & Moè (2014)* | 8th grade students | The play fighting intervention consisted in a progression of games and exercises, implicating touch, physical contact and opposition | Standard volleyball lessons | 4 weeks, 8 lessons, 2 times a week | [abc] |
| RCT | Int. $n = 103$ | | | | |
| Italy | Con. $n = 107$ | | | | |
| | $13.27 \pm 0.48$ years | | | | |
| | 42% girls | | | | |
| *Reynes & Lorant (2002)* | Primary school boys | Judo practice | No training | 1 year, 2 sessions per week | [abc] |
| Quasi-experimental Study | $n = 55$ | | | | |
| France | Int. $n = 28$ | | | | |
| | Con. $n = 27$ | | | | |
| | 8 years | | | | |

**Notes.**

Bullets are used to distinguish between the two different intervention methods.

[a] Aggression.

[b] Hostility.

[c] Anger.

[d] Delinquent Acts.

[e] Attitude Towards Violence.

[f] Provocation/Bullying.

Yang et al. (2023), *PeerJ*, DOI 10.7717/peerj.15504

**Table 2  Risk of bias from quasi-experimental studies.**

| Study | Is it clear in the study what is the 'cause' and what is the 'effect'? | Were the participants included in any comparisons similar? | Were the participants included in any comparisons receiving similar treatment/ care, other than the exposure or intervention of interest? | Was there a control group? | Were there multiple measurements of the outcome both pre and post the intervention/ exposure? | Was follow-up complete, and if not, was follow-up adequately reported and strategies to deal with loss to follow-up employed? | Were the outcomes of participants included in any comparisons measured in the same way? | Were outcomes measured in a reliable way? | Was appropriate statistical analysis used? |
|---|---|---|---|---|---|---|---|---|---|
| *Mehralian et al. (2022)* | Y | U | Y | Y | N | Y | Y | Y | Y |
| *Rosa et al. (2021)* | Y | Y | Y | N | N | Y | Y | Y | Y |
| *Harwood-Gross et al. (2021)* | Y | Y | N | Y | N | Y | Y | Y | Y |
| *Blomqvist (2020)* | Y | Y | U | N | Y | Y | Y | Y | Y |
| *Hortiguela, Gutierrez-Garcia & Hernando-Garijo (2017)* | Y | Y | N | Y | N | U | Y | Y | Y |
| *Park et al. (2017)* | Y | Y | Y | Y | N | Y | Y | Y | Y |
| *Pels & Kleinert (2016)* | Y | U | Y | N | N | N/A | Y | Y | Y |
| *Shachar et al. (2016)* | Y | Y | U | Y | Y | Y | Y | Y | Y |
| *Reynes & Lorant (2002)* | Y | Y | Y | Y | N | Y | Y | Y | Y |

**Notes.**

Y, Yes; N, No; U, Unclear; N/A, Not applicable.

**Table 3  Risk of bias from RCTs.**

| Study | Selection bias | | Performance bias | Detection bias | Attrition bias | Reporting bias | Other bias |
|---|---|---|---|---|---|---|---|
| | Random sequence generation | Allocation concealment | Blinding of participants and personnel | Blinding of outcome assessment | Incomplete outcome data | Selective reporting | |
| Trajković et al. (2020a) | Unclear | Unclear | Unclear | Unclear | Unclear | Unclear | Unclear |
| Trajković et al. (2020b) | Unclear | Unclear | Unclear | Unclear | Low risk | Low risk | Unclear |
| Wade et al. (2018) | Unclear | Unclear | Unclear | Unclear | Low risk | Low risk | Unclear |
| Fung & Lee (2018) | Low risk | Low risk | Low risk | Unclear | Low risk | Low risk | Unclear |
| Setty, Subramanya & Mahadevan (2017) | Unclear | Unclear | Unclear | Unclear | Low risk | Low risk | Unclear |
| Carraro, Gobbi & Moè (2014) | Unclear | Unclear | Unclear | Unclear | Low risk | Low risk | Unclear |

**Table 4  Subgroup analysis of aggression.**

| Study characteristics | Number of studies (sample size) | SMD | 95% Cl | P | $I^2$ |
|---|---|---|---|---|---|
| Type of sport | | | | | |
| Non-contact sport | 5 (1123) | −0.65 | −1.17, −0.13 | 0.020 | 92% |
| High-contact sport | 4 (509) | −0.15 | −0.55, 0.25 | 0.470 | 79% |
| Intervention duration | | | | | |
| <6 months | 4 (427) | −0.99 | −1.73, −0.26 | 0.008 | 90% |
| ≥6 months | 5 (1205) | −0.08 | −0.44, 0.28 | 0.660 | 87% |

*Harwood-Gross et al. (2021)* only provided the mean change-score comparison between martial arts training and controls on aggression so that it was not included in the meta-analysis. The aggression scores in both groups were increased, and the difference was not significant ($P = 0.85$).

*Blomqvist (2020)* compared effects of Mixed Martial Arts (MMA) intervention and Brazilian Jiu-Jitsu (BJJ) intervention on aggression. The results showed that there was no significant main effect of aggression as a result of training ($P = 0.100$). However, the interaction between aggression and sport was significant ($P < 0.001$). Whereas MMA practitioners slightly increased their levels of aggression, BJJ practitioners reduced theirs. *Pels & Kleinert (2016)* reported a significant reduction of aggressive feelings was found for participants exercising individually in the rowing condition compared with the individual combat exercise condition.

## Impact of interventions on hostility

The overall mean effect size of five included studies (*Trajković et al., 2020a*; *Trajković et al., 2020b*; *Shachar et al., 2016*; *Carraro, Gobbi & Moè, 2014*; *Reynes & Lorant, 2002*) indicated

**Table 5  Sensitivity analysis of aggression, hostility and anger.**

| Removed study | MD (95% Cl) | P | $I^2$ |
|---|---|---|---|
| Aggression | | | |
| *Mehralian et al. (2022)* | −0.23 (−0.48, 0.03) | 0.080 | 81% |
| *Trajković et al. (2020a)* | −0.44 (−0.78, −0.11) | 0.010 | 80% |
| *Trajković et al. (2020b)* | −0.40 (−0.76, −0.05) | 0.030 | 89% |
| *Wade et al. (2018)* | −0.43 (−0.80, −0.07) | 0.020 | 88% |
| *Fung & Lee (2018)* | −0.42 (−0.77, −0.06) | 0.020 | 89% |
| *Park et al. (2017)* | −0.34 (−0.68, 0.00) | 0.050 | 89% |
| *Shachar et al. (2016)* | −0.36 (−0.74, 0.02) | 0.060 | 87% |
| *Carraro, Gobbi & Moè (2014)* | −0.36 (−0.72, 0.01) | 0.060 | 89% |
| *Reynes & Lorant (2002)* | −0.46 (−0.78, −0.15) | 0.004 | 87% |
| Hostility | | | |
| *Trajković et al. (2020a)* | −0.30 (−0.42, −0.17) | <0.001 | 9% |
| *Trajković et al. (2020b)* | −0.30 (−0.42, −0.18) | <0.001 | 5% |
| *Shachar et al. (2016)* | −0.23 (−0.41, −0.05) | 0.010 | 0% |
| *Carraro, Gobbi & Moè (2014)* | −0.27 (−0.40, −0.14) | <0.001 | 5% |
| *Reynes & Lorant (2002)* | −0.31 (−0.43, −0.19) | <0.001 | 0% |
| Anger | | | |
| *Trajković et al. (2020a)* | −0.19 (−0.56, 0.17) | 0.300 | 83% |
| *Trajković et al. (2020b)* | −0.20 (−0.57, 0.17) | 0.290 | 83% |
| *Shachar et al. (2016)* | −0.18 (−0.62, 0.26) | 0.430 | 81% |
| *Carraro, Gobbi & Moè (2014)* | −0.23 (−0.65, 0.19) | 0.280 | 83% |
| *Reynes & Lorant (2002)* | −0.43 (−0.55, −0.31) | <0.001 | 0% |

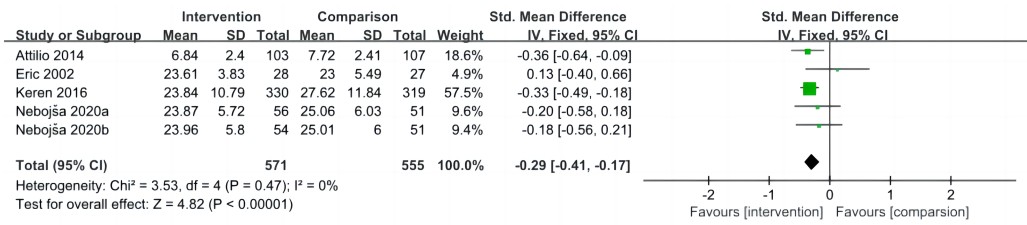

**Figure 3  Forest plot of studies for hostility.**

that sport interventions was associated with lower hostility (SMD = −0.29, 95% CI [−0.41 to −0.17], $P < 0.001$; $I^2 = 0\%$) (Fig. 3).

When the impact of individual studies was examined by removing studies from the analysis one at a time, we observed that the pooled results estimate remained consistent.

## Impact of interventions on anger

The overall mean effect size of five included studies (*Trajković et al., 2020a*; *Trajković et al., 2020b*; *Shachar et al., 2016*; *Carraro, Gobbi & Moè, 2014*; *Reynes & Lorant, 2002*) indicated

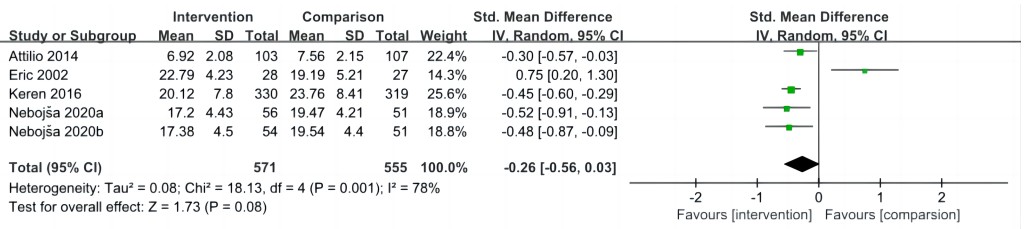

**Figure 4   Forest plot of studies for anger.**

that sport interventions was not associated with lower anger (SMD = −0.26, 95% CI [−0.56 to 0.03], $P = 0.08$; $I^2 = 78\%$) (Fig. 4).

Sensitivity analysis showed that when *Reynes & Lorant (2002)* been removed, the heterogeneity became small ($I^2 = 0\%$) and the pooled result became significant ($P < 0.001$) (Table 5).

## Impact of interventions on delinquent acts

*Blomqvist (2020)* indicated that both MMA and BJJ intervention groups reduced criminal behaviour moderately ($P = 0.030$). *Fung & Lee (2018)* found that Chinese martial arts group had light decrease in delinquent behavior than did the physical fitness training group, but there was no significant fixed effects of training were found in delinquent behavior ($P = 0.760$).

## Impact of interventions on attitude towards violence

*Setty, Subramanya & Mahadevan (2017)* and *Hortiguela, Gutierrez-Garcia & Hernando-Garijo (2017)* reported effects of sport intervention on adolescents' attitude towards violence. *Setty, Subramanya & Mahadevan (2017)* showed a significant change in both yoga and control groups in self-reported ATV, pre- and post-intervention ($p < 0.05$). But the mean change in the yoga group is 39.59%, compared to 7.51% in the control group, indicating significant improvement. *Hortiguela, Gutierrez-Garcia & Hernando-Garijo (2017)* reported the results of two dimensions of ATV—unjustified violence and violence linked to self protection. The unjustified violence and the violence linked to self protection fell from high to medium in the judo and capoeira teaching units, significant difference with large effect sizes were found between the pre-test and the pos $t$-test in unjustified violence ($P = 0.021$) while there was no difference in the control group.

## Impact of interventions on provocation/bullying

*Rosa et al. (2021)* carried out judo and ball games among children and adolecents. A significant improvement in the domain of provocation/bullying was observed after the interventions, with judo increasing 18.1% and ball games increasing 4.1%. In other words, the participants felt safer and more confident about other people's negative attitudes.

## DISCUSSION

This review evaluated the effectiveness of existing sports interventions to reduce aggression in children and adolescents. The overall results showed that sports intervention could reduce the aggression and hostility of children and adolescents and could not reduce the anger, while the evidence is indeterminate at the domain level for delinquent acts, attitude towards violence and provocation/bullying.

A strong relationship between sport and aggression has been reported in the literature. *Pino-Juste, Portela-Pino & Soto-Carballo (2019)* reported that the higher the index of physical activity is, the lower the level of aggressiveness is. A systematic review pointed out that physical education played an important role in the prevention of bullying (*Jimenez-Barbero et al., 2020*). Another review found that positive youth development Interventions with a physical activity component among pre- and early adolescents aged 8–14 years may lower bullying behaviors (*Majed et al., 2022*). *Gråstén & Yli-Piipari (2019)* indicated that violence among children and bullying reduced during the Physical Activity as Civil Skill Program according to teachers' written feedback. These are consistent with our results. *The European Commission's White Paper on Sport (2007)* pointed out that the social code implied in the sports include fair competition and team spirit, which can cultivate teenagers' social behavior patterns and reduce their aggressive behavior. *Lorenz (1966)* believed sport was a ritualized venting of aggression, which teaches people to consciously and responsibly control their fighting behavior. However, the results of sensitivity analysis indicated the lack of robustness of the meta-analysis. This may be because the small sample sizes of the studies included in the meta-analysis and different basic characteristics of the studies led to a large heterogeneity of the pooled results, which requires cautious interpretation.

Subgroup analyses showed that non-contact sports were associated with lower aggression while high-contact sports were not. *Sofia & Cruz (2017)* surveyed 141 athletes from different types of sport and found the same result: athletes from sports with higher levels of physical contact tended to be more aggressive than those from sports with lower levels of contact. This may be because self-control lies in the central role in the regulation of aggression in sport (*Sofia & Cruz, 2015*; *Sofia & Cruz, 2016*). High-contact sports mean strong competition and impulsivity. People who participate in non-contact sports could better control their aggressive behavior. In addition, the comparison between rowing and combat exercise also confirmed this opinion (*Pels & Kleinert, 2016*), the non-contact rowing can reduce aggression more than the high-contact combat.

Moreover, when the intervention duration ≥6 months, sport interventions was not associated with lower aggression. There is no study focusing on the influence of the duration of sport intervention on the effect of intervention currently. *Stansfield (2017)* confirmed that higher levels of participation in sports increased violence involvement. *Méndez, Ruiz-Esteban & Ortega (2019)* also indicated that students with high exercise frequency were more aggressive than those with low exercise frequency. Due to the large difference of intervention frequency among the included studies, we did not conduct subgroup analysis. According to the result, it could conceivably be hypothesised that whether the low level of sport involvement releases the aggressive impulse, and with the

accumulation of exercise, the aggressive impulse rises again. As we all know, high level of exercise can improve muscle strength, and muscular strength may be an important predictor of aggression in bullying (*Benítez-Sillero et al., 2021*), this view also supported our hypothesis.

From the above mentioned, we have reason to believe that low level of non-contact sports involvement may be more conducive to the release of aggressive impulses, so then reducing aggressiveness of children and adolescents. Nevertheless, what kind of sport intervention frequency and duration can play the largest role in it needs further research.

## LIMITATION

One limitation of this review was that the intervention programme of included studies were highly diversified with small sizes and varying assessment methods, which resulted in the high heterogeneity. Highly diverse sport intervention programs suggested that there might be other variables besides sport, such as the attention given to the participants, the approach of the coaches, the 'winning at all costs' philosophy of sport and cultures of different countries, that could also play a role in the results. Further studies are necessary to make clear which variables are actually factors contributing to aggression. Another limitation was that some of the included studies could not be meta-analysed due to the lack of standard control groups or the inability to extract data, so only descriptive statistics were made. Besides, the included studies were limited to peer-reviewed journals in English identified by the search strategy, potentially omitting other relevant studies.

## CONCLUSION

This review confirmed that sports intervention can reduce the aggression of children and adolescents. We suggested that schools can organize young people to participate in low-level, non-contact sports to reduce the occurrence of bullying, violence and other aggression-related adverse events. Additional studies are needed to determine which other variables are associated with aggression in children and adolescents, in order to develop a more detailed and comprehensive intervention programme to reduce their aggression.

## ACKNOWLEDGEMENTS

We are grateful to Yangzhou University for providing the databases.

### Funding

Fengshu Zhu was supported by the National Social Science Foundation of China: Research on the Construction and Application of Targeted Sport Intervention Model for Adolescent Aggressive Behavior (20BTY118). The funders had no role in study design, data collection and analysis, decision to publish, or preparation of the manuscript.

## Grant Disclosures

The following grant information was disclosed by the authors:

National Social Science Foundation of China: Research on the Construction and Application of Targeted Sport Intervention Model for Adolescent Aggressive Behavior: 20BTY118.

## Competing Interests

The authors declare there are no competing interests.

## Author Contributions

- Yahui Yang conceived and designed the experiments, performed the experiments, analyzed the data, prepared figures and/or tables, authored or reviewed drafts of the article, and approved the final draft.
- Hao Zhu performed the experiments, prepared figures and/or tables, selection of studies, extraction of data and assessment of risk of bias, and approved the final draft.
- Kequn Chu performed the experiments, prepared figures and/or tables, selection of studies and extraction of data, and approved the final draft.
- Yue Zheng analyzed the data, prepared figures and/or tables, and approved the final draft.
- Fengshu Zhu conceived and designed the experiments, authored or reviewed drafts of the article, and approved the final draft.

## Data Availability

This study is a systematic review and meta-analysis, using data obtained from the included studies rather than raw data. Studies included in this analysis are available in Table 1.

## Supplemental Information

Supplemental information for this article can be found online at http://dx.doi.org/10.7717/peerj.15504#supplemental-information.

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
