# Peer review of "Effects of sports intervention on aggression in children and adolescents: a systematic review and meta-analysis"

_PeerJ, doi:10.7717/peerj.15504_

## Round 0.1 · original submission · Major Revisions

Dear Authors,

Please revise according to the suggestions of the reviewers or write a detailed rebuttal on a point-by-point basis.

·

Basic reporting

We suggest a proofreading by a fluent English speaker.

Also, it is found that the approach to the sport has a crucial role in it's effect. This doesn't necessarily has to be a high-contact sport : "A "win-at-all-costs" philosophy has often led to unethical and aggressive behaviors, impacting negatively and destructively on the development and well being of young athletes and of society at large. Researchers (e.g.; Arms, Russell, & Sandilands, 1979; Bredemeier, Weiss, Shields, & Cooper, 1986; Ewing, Gano-Overway, Branta, & Seefeldt, 2002; Guivernau & Duda, 2002; Terry & Jackson, 1985)"

We suggest more details and information should be included. Since there was no association between sport interventions and lower aggression after the period of 6 months, it is questionable if the effect is the result of sport interventions or the effect of attention given to the sportsmen as well as other influences of the process. Also, different sports of the same type, for instance mma vs brasilian jiu jitsu ( jiu jitsu doesn't teach stiking), may have different results on aggression, which begs the question of the impact of the philosophy of a specific sport. If sport itself is not associated with lower levels of anger, we can not exclude that the philosophy, the impact of teaching and attention payed may still play an important role. We could suggest that sport interventions may be used along with psychosocial support to have a more positive effects.

Experimental design

We suggest more information about the limitations such as the fact sports was looked at as one variable. Some studies mentioned suggest different conclusions. Since there are many types of very different sports that fit into the same category as well as the approach of the coaches (it there was always one present) which may have a role in the effect of the mentioned variable, this may play a role in the results.

Validity of the findings

The review only suggests the possibility of sports interventions reducing aggression, but since the limitations of the studies used in the pool are significant and the variables aren't clear since every study had it's own variables, we suggest the conclusion also notes that further studies are necessary to make clear which variables are actually the ones causing the effect. No cultural aspects were addressed, which may play a significant role. Also no effect of the intervention itself as a form of attention was addressed.

Reviewer 2 ·

Basic reporting

The study was well-designed, and the authors clearly described the literature search criteria, definition outcomes, and search process. Two comments:

What type of studies included in the analysis, cohort, case-control, or any other type of studies? The study design of the included studies may impact the interpretation of the study results.

The study results show high I^2, which means considerable heterogeneity. Given that, are the results reliable from the analysis? The authors touched on this topic in the limitation a little bit, but did not discuss the potential impact of the study result interpretation.

Experimental design

NA

Validity of the findings

NA

Reviewer 3 ·

Basic reporting

In this manuscript, the authors review articles that investigate the effect of sports characteristics on the aggression behavior of children. They deem 15 articles, all in English, to be suitable for this review.

I think this is a reasonable review article and can be accepted with minor revisions:

1. Line 125: Karen et a -> Karen et al

2. Line 165 and several other places have mentions of delinquent acts. Please describe or give an example of these acts

3. Line 167: .... carried out an single experiment..... -> carried out a single experiment

4. Line 259: ... sport was a ritualize venting....-> sport was a ritualized venting

Experimental design

The description of the Methods is lacking in this manuscript. I am unclear on what the I^2 means as reported in the text in several lines, e.g., Line 178. Please add the description of the methods used more clearly

Validity of the findings

I believe the findings are valid but the methodology needs to be more descriptive. See my comment above

---

## Round 0.2 · accepted · Accept

Dear Authors,

Your manuscript is now acceptable for publication in its current form.

·

Basic reporting

NO COMMENT

Experimental design

no comment

Validity of the findings

no comment

Additional comments

no comment

Reviewer 2 ·

Basic reporting

The authors have addressed my comments appropriately. I have no more comments.

Experimental design

The authors have addressed my comments appropriately. I have no more comments.

Validity of the findings

The authors have addressed my comments appropriately. I have no more comments.